# Oil spill model uncertainty quantification using an atmospheric ensemble

Konstantinos Kampouris[1], Vassilios Vervatis[1], John Karagiorgos[1], Sarantis Sofianos[1]

[1]University of Athens, Department of Physics, Athens, Greece

*Correspondence to*: Konstantinos Kampouris (kkampour@uoa.gr)

**Abstract.** We investigate the impact of atmospheric forcing uncertainties on the prediction of dispersion of pollutants in the marine environment. Ensemble simulations consisted of 50 members were carried out using the ECMWF ensemble prediction system and the oil spill model MEDSLIK-II in the Aegean Sea. A deterministic control run, using the unperturbed wind of the ECMWF high resolution system, served as reference for the oil spill prediction. We considered oil spill rates and duration

similar to major accidents of the past (e.g. the Prestige case) and we performed simulations for different seasons and oil spill types. Oil spill performance metrics and indices were introduced in the context of probabilistic hazard assessment. Results suggest that oil spill model uncertainties were sensitive to the atmospheric forcing uncertainties, especially to phase differences in the intensity and direction of the wind among members. An oil spill ensemble prediction system based on model uncertainty of the atmospheric forcing, shows great potential for predicting pathways of oil spill transport, alongside a deterministic

simulation, increasing the reliability of the model prediction and providing important information for the control and mitigation strategies in the event of an oil spill accident.

## 1 Introduction

Although unintentional oil pollution caused by ships is declining over the years, increased oil shipments may pose an increased risk. In the event of an oil spill accident, oil spill model predictions serve as the forefront tools to assist regional and national

contingency plans (Zodiatis et al., 2017a). The behaviour of some environmental variables may alter the physical and chemical processes acting on oil spills (Zodiatis et al., 2017b). Uncertainties related to parameters like metocean conditions, influence the transport and weathering of oil and the accuracy of oil spill model predictions. The identification of such factors, their sensitivity and the evaluation of models are necessary for improving oil spill forecasting.

The wind is a major source of errors in an oil spill modelling (Li et al., 2013, 2019; Khade et al., 2017). Incomplete knowledge

of atmospheric initial conditions and simplifications in atmospheric model parameterizations due to constrains in computational resources, are major sources of uncertainty in numerical weather prediction systems (Buizza, 2016). A method to take under account atmospheric model errors and improve oil spill model prediction is to follow an ensemble-based approach, using different forecasts, as opposed to a single deterministic run. An ensemble of forecasts is represented by a number of different, but equally possible model states, generated by perturbed initial conditions and state variables. The

ensemble spread can be used as a proxy of model errors in the forecast. A large spread increases the possibility some of the ensemble forecasts to be closer to the observed oil spill state. Ensemble simulations have been used in the past to assess the risk of oil spills and their potential environmental impact, considering major sources of uncertainties, like the oil release positions, the oil characteristics, and the metocean conditions during the accident. For example, ensemble oil spill simulations have been used for hazard and risk assessment by Price et al. (2003), Goldman et al. (2015), Liubartseva et al. (2015, 2016), Jiménez Madrid et al. (2016), Al Shami et al. (2017), Olita et al. (2019), Amir-Heidari and Raie (2019) and Sepp Neves et al. (2015, 2016, 2020). Mariano et al. (2011) performed an ensemble to assess uncertainties in the oil spill state and spreading. Perturbed forcing fields have been used to assess their impact on an oil spill forecasting system by Jorda et al. (2007), and stochastic methods have been applied on the transport and oil spill transformations by Snow et al. (2014) and Rutherford et al. (2015). Khade et al. (2017) investigated the potential of atmospheric ensemble forecasting on the Deep Water Horizon oil spill accident in the Gulf of Mexico.

Maritime transport is a major source of pollution from oil and polycyclic aromatic hydrocarbons in the Mediterranean Sea, and it has been shown that the distribution of oil spills is associated with major shipping routes (UNEP/MAP, 2012). The total activity of vessels in the Mediterranean has been steadily increasing in recent years and is expected to continue over the next decade. Large merchant vessels increasingly operate in the Mediterranean Sea to transport goods. As a result, operational systems have been developed to assess the risk of oil spills in areas with high-density vessel traffic (Quattrocchi et al., 2021). The main oil transport route (90% of the total traffic), extends from the eastern to the western Mediterranean and connects the passages of the Dardanelles Strait and the Suez Canal with the Straits of Gibraltar (UNEP/MAP, 2012). The Aegean Sea, in particular, as one of the world's busiest waterways, shows a relatively high risk for oil spills, having one of the highest numbers of maritime accidents in relation to other areas in the Mediterranean Sea (EMSA, 2019). Also, it is a basin with complex bathymetry and coastline, including intense weather phenomena and ocean circulation patterns with strong seasonality. For all these reasons, the implementation of an oil spill probabilistic system in the region, using as information an ensemble of wind forcing uncertainties, is of great interest.

The study aims at assessing the impact of atmospheric forcing uncertainties on the model prediction of oil spill and dispersion of pollutants in the marine environment. We used an ensemble-based approach for the simulation of oil spill in a regional domain for the Aegean Sea. The model incorporated wind forcing from the ECMWF ensemble prediction system, generating an ensemble of oil spill forecasts. Ensemble-based metrics and indices were introduced to answer if the ensemble of oil spill forecasts can provide additional information with respect to a deterministic simulation, providing the decision-makers with several equally possible outcomes, to better plan mitigation procedures. The experimental setup and the ensemble-based metrics are presented in Section 2. The oil spill results are presented in Section 3 and the conclusions in Section 4.

## 2 Methodology

### 2.1 Oil spill model

The numerical model MEDSLIK-II (De Dominicis et al., 2013a, 2013b), is a freely available community model, based on its precursor MEDSLIK (Lardner et al., 1998, 2006; Zodiatis et al., 2005, 2008). It is designed to predict the transport and weathering of an oil spill, caused by complex physical processes occurring at the sea surface, using a Lagrangian representation of the oil slick. This numerical representation requires the following different state variables: the oil slick, the particle and the structural state variables, which are all used for different calculations. The transformation and movement of an oil slick depend on many factors, the main ones being: the meteorological and oceanographic conditions at the air-sea interface, the wind forcing and marine currents in the oil spill area, and the chemical characteristics of the oil, the initial volume and the rate of oil release.

A brief description of the basic equations used by MEDSLIK-II is given below, following De Dominicis (2012), De Dominicis et al. (2013a, 2013b), and Liubartseva et al. (2020). The oil spill concentration changes over time due to physical and chemical processes, also known collectively as "weathering", e.g., evaporation, emulsification, dispersion in the water column, and spreading. The general active tracer equation for oil in a marine environment is:

$$\frac{\partial C}{\partial t} + \boldsymbol{U} \cdot \nabla C = \nabla \cdot (\boldsymbol{K} \nabla C) + \sum_{j=1}^{M} r_j(C) \ , \tag{1}$$

where $C$ is the total oil concentration with units of mass over volume (kg/m$^3$), $\partial/\partial t$ is the local time-rate-of-change operator, $\boldsymbol{U}$ is the sea current mean field (including also wind-wave properties in the sea surface), $\boldsymbol{K}$ is the turbulent diffusivity tensor, and $r_j(C)$ are the $j = 1, \dots, M$ transformation rates that modify the tracer concentration due to physical and chemical transformation processes. The equation (Eq. 1) is divided into two components:

$$\frac{\partial C_1}{\partial t} = \sum_{j=1}^{M} r_j(C_1) \ , \tag{2}$$

$$\frac{\partial C}{\partial t} = -\mathbf{U} \cdot \nabla C_1 + \nabla \cdot (\boldsymbol{K} \nabla C_1) \ , \tag{3}$$

first, the weathering transformation equation (Eq. 2), where $C_1$ is the concentration of oil considering only the weathering processes, and second, the Lagrangian advection-diffusion equation (Eq. 3), discretizing the oil slick into a large number of particles (with associated particle state variables), transported by advection and diffusion processes. The transformation processes, calculated using the Mackay et al. (1980) fate algorithms, act on the total volume of the oil slick. The surface volume of the oil slick is classified into a thin part at the edges of the oil slick, and a thick part near its center. Weathering occurs on the sea surface oil and comprises of three main processes, i.e. evaporation, dispersion and spreading. The total concentration $C$ is classified into structural state variables, i.e. oil concentrations at the surface, the subsurface, and oil adsorbed on the sea shore and in bottom sediments. The weathering transformation Eq. (2) is solved calculating the concentration $C_1$, which is then used by the advection-diffusion Eq. (3) calculating the total concentration $C$.

The oceanic and atmospheric forcing fields for the oil spill model, are used to calculate the change of the oil spill particle positions, with the mean field $\boldsymbol{U}$ in Eqs. (1) and (3) to be a sum of different components; in the sea surface:

$$\boldsymbol{U}|_{z=0} = \boldsymbol{U}_C + \boldsymbol{U}_W + \boldsymbol{U}_S + \boldsymbol{U}_D \, , \tag{4}$$

and in the water column:

$$\boldsymbol{U} = \boldsymbol{U}_C \, , \tag{5}$$

$\boldsymbol{U}_C$ is the forcing input Eulerian field for the sea current velocity term, $\boldsymbol{U}_W$ is the local wind velocity correction term, due to uncertainties in simulating the Ekman transport pattern parameterized as a function of wind intensity and angle between winds and currents, $\boldsymbol{U}_S$ is the velocity of wave-induced currents due to Stokes drift calculated by the oil spill model and $\boldsymbol{U}_D$ is a wind drag correction due to emergent part of the objects at the sea surface. In our study, the oil spill model uncertainties are attributable to the different wind forcing per member derived from the ECMWF ensemble prediction system, and consequently 100 to the different correction terms in Eq. (4), i.e. the Ekman transport correction $\boldsymbol{U}_W$, the wind drag correction $\boldsymbol{U}_D$ and the wave-induced Stokes drift $\boldsymbol{U}_S$.

## 2.2 Ensemble experiment setup

The modelling area focuses on the Aegean Sea including the Kafireas Strait which is one of the main traffic routes in the Mediterranean, especially for the transportation of crude oil from the Black Sea. The model domain encompasses islands and 105 islets over the Cyclades plateau, with complex bathymetry and coastlines. Figure 1 shows the bathymetry and coastline data used in MEDSLIK-II simulations, along with the names of the geographic locations and the release location of the oil spill particles. The bathymetry is based on the General Bathymetric Chart of the Oceans (GEBCO_2014; Weatherall et al., 2015), delivered on a global grid at 30 arc-second intervals and the oil spill model domain spans the area from 23 ° E to 26 ° E and 36 ° N to 39 ° N. For the coastlines, we used the version 2.3.7 of the high-resolution GSHHG geographic dataset (Wessel and 110 Smith, 1996). A characteristic feature of the ocean circulation is the strong currents exchanged through the Kafireas Strait, with the potential to lead in an extensive pollution in the area in case of an oil spill accident.

We performed an ensemble of 50 numerical simulations where each oil spill member uses different atmospheric forcing, obtained from an atmospheric ensemble, to assess the impact of wind forcing uncertainties on the performance of the oil spill model to predict the transport of pollutants in the marine environment. We also performed an oil spill simulation using a 115 deterministic atmospheric forcing as a reference in the assessment of our results. Different accident scenarios of oil spill simulations were considered for two different seasons (i.e. winter and spring) and for three oil types. A single oil release station was chosen at Kafireas Strait (Fig. 1), performing 7-day forecast simulations and continuous oil release with a spillage rate of 5 tons per hour. The setup of the oil spill duration and rate were chosen according to major accidents of the past, for example here the Prestige case (Portman, 2016; Sepp Neves et al., 2016). The number of parcels used in the simulations was a total of 120 $10^5$, to estimate dispersion and oil slick concentrations. The horizontal and vertical diffusion coefficients remained constant

throughout the simulation, using the default MEDSLIK-II values. The oil spill model estimates wind-wave corrections based on Ekman transport and Stokes drift, taking also into account the mixed layer depth for the different periods in January and May 2017, at 50 m and 10 m respectively. The run time of our simulations was mainly determined by the number of oil parcels and the size of the ensemble. For a 168-hour (7-day) oil spill prediction (in our domain of interest depicted in Fig. 1), the deterministic simulation required approximately a 20-minute run time, including the model's I/O tasks. The computational cost of the ensemble prediction, in case all members are run in sequence (i.e., one after the other), is analogous to the number of the ensemble members. The latter is valid also for the data storage. For a small ensemble simulation in a HPC facility with available CPU cores, the ensemble members can run simultaneously and the computational cost is the same as the deterministic simulation.

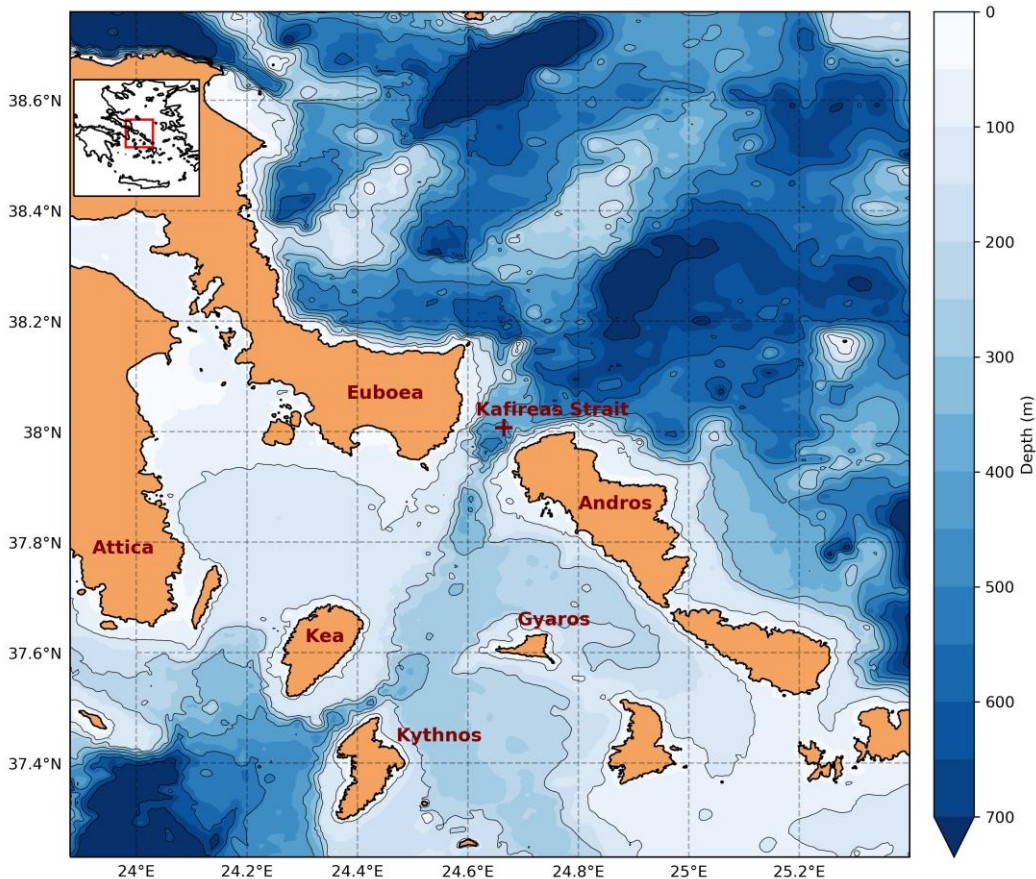

**Figure 1: Aegean Sea bathymetry in meters (GEBCO_2014; Weatherall et al., 2015) and coastlines (Wessel and Smith, 1996) of the study area, and oil spill release location at Kafireas Strait (red cross).**

Additional experimental options for the initial, boundary and forcing conditions in the accident scenarios were the following: (a) the ECMWF high resolution deterministic forcing at ~9km resolution and (b) the ECMWF ensemble prediction system of 50 members at ~18km resolution, (c) the ocean analysis of current velocities and temperature retrieved by the CMEMS

infrastructure (https://doi.org/10.25423/CMCC/MEDSEA_ANALYSIS_FORECAST_PHY_006_013_EAS4) with horizontal resolution ~4 km and depths at 0 m, 10 m, 30 m and 120 m, and finally, (d) three types of oil with API 12, API 31 and API 38, representing heavier, medium and lighter oil spills over a wide range of oil densities (Sepp Neves et al., 2016). The wind and oceanic forcing fields used in our experiments were in a format supported by MEDSLIK-II and the datasets retrieved by

CMEMS and ECMWF archives were preprocessed with tools available by the oil spill platform (e.g. converting the oil spill model inputs from the CMEMS daily ocean analysis and the ECMWF 3-hour atmospheric forcing to hourly fields).

Figure 2 shows rose diagrams of sea surface current velocities, during the two periods under investigation and at the location of the oil spill particles release (red cross on Fig. 1). Fig. 2a shows the velocities of the surface sea current during winter from 2017-01-10 to 2017-01-16 and Fig. 2b during spring from 2017-05-10 to 2017-05-16. The prevailing direction of the sea

surface current is to the south-southwest during both periods, which is the main circulation pattern at the Kafireas Strait, with relatively high velocities reaching up to 0.4 m/s in winter and 0.5 m/s in spring.

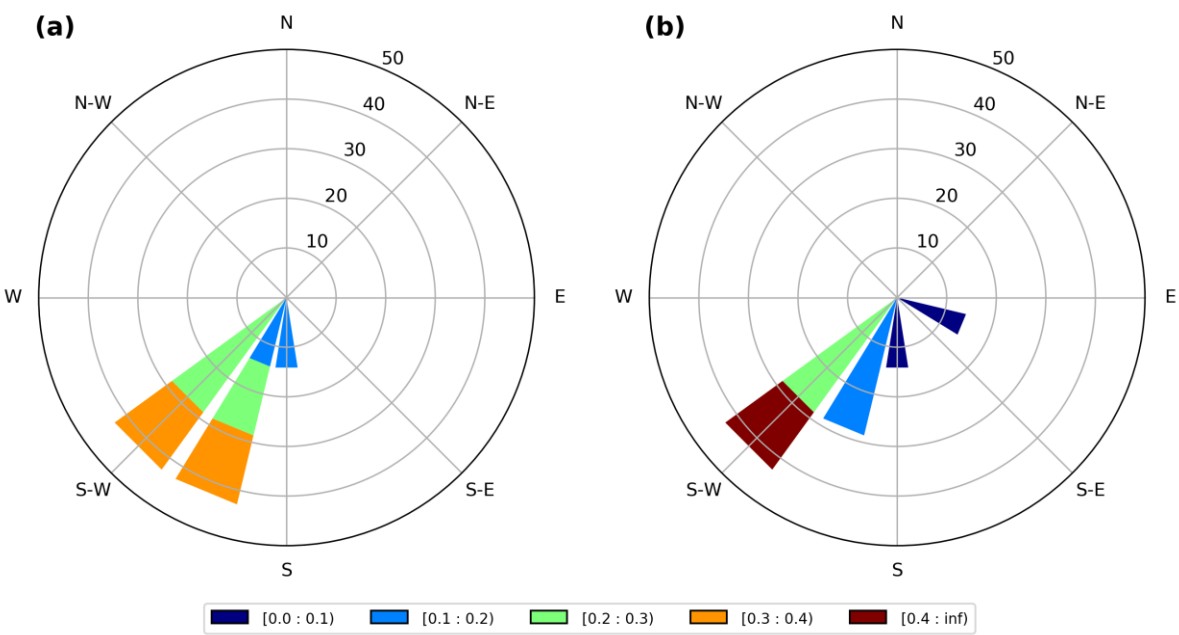

**Figure 2: Current roses of sea surface velocities at Kafireas Strait: (a) winter and (b) spring. Colours in units (m/s) and iso-contours (%).**

Figure 3 shows the wind roses of the atmospheric forcing at the release location of the oil spill in Kafireas Strait, quantitatively assessing in terms of percentages, wind speed and direction of the prevailing wind patterns. Fig. 3a shows the wind velocities and directions of the deterministic simulation for the winter period in January 10-16, 2017, and Fig. 3c shows all 50 members of the atmospheric ensemble for the same period. The prevailing wind direction is to the north-northeast, nearly opposite to the sea currents in the area, with maximum wind speed values exceeding 10 m/s. The wind of the ensemble shows larger

variability compared with the deterministic forcing, denoting an ensemble spread in wind speed and direction. Similarly, Fig.

3b and Fig. 3d, show the wind velocities of the deterministic simulation and the ensemble respectively, for the spring period in May 10-16, 2017. The prevailing wind direction is to the south, with a maximum value up to 10 m/s. During spring, the intensity of the wind is in general lower than in winter and the prevailing wind direction is similar to that of the sea currents in the area. Overall, the differences between the deterministic and the ensemble atmospheric forcings are smaller than those during winter.

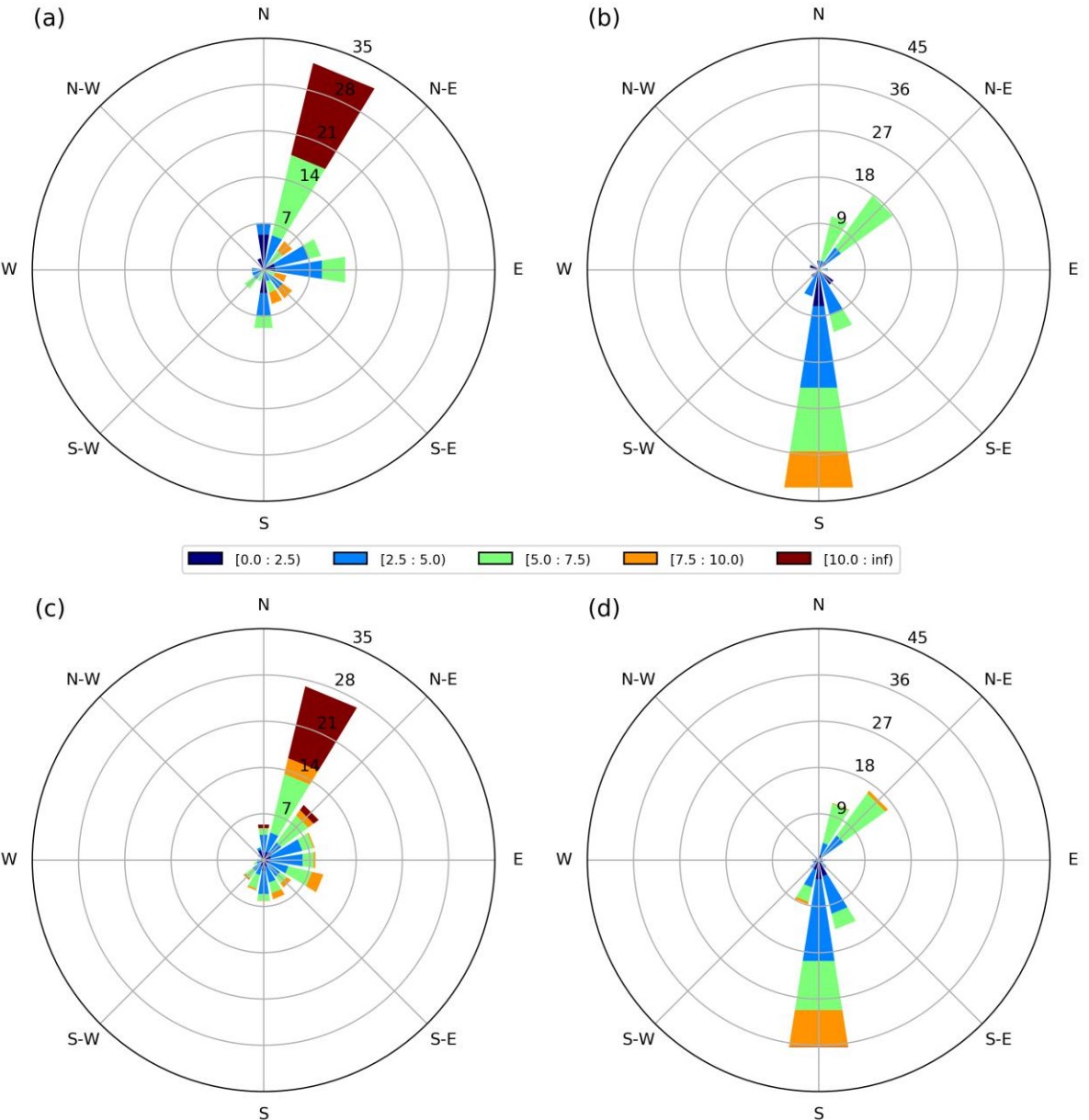

Figure 3: Wind roses of velocity and direction at Kafireas Strait, for the 7-day deterministic simulations during (a) winter and (b) spring; (c-d) same for 50 ensemble members. Colours in units (m/s) and iso-contours (%).

In Fig. 4, we show the wind variations at the release location of the oil spill, for the 7-day simulation period during winter and spring, and for the deterministic and the ensemble members. The wind vector plots indicate that there are both gradual and abrupt changes in wind speed and direction, showing larger variability during winter than spring. Wind forcing uncertainties are attributed (1) to phase errors during transient changes in wind direction between the deterministic and the ensemble members, and (2) to wind speed uncertainties mostly for the less intense winds. For instance, during winter, there are abrupt changes in wind speed and direction in the middle and at the end of the run, showing that the ensemble members may differ significantly with respect to the deterministic state (e.g. having members with opposite wind directions and few hours' time-lagged between one another).

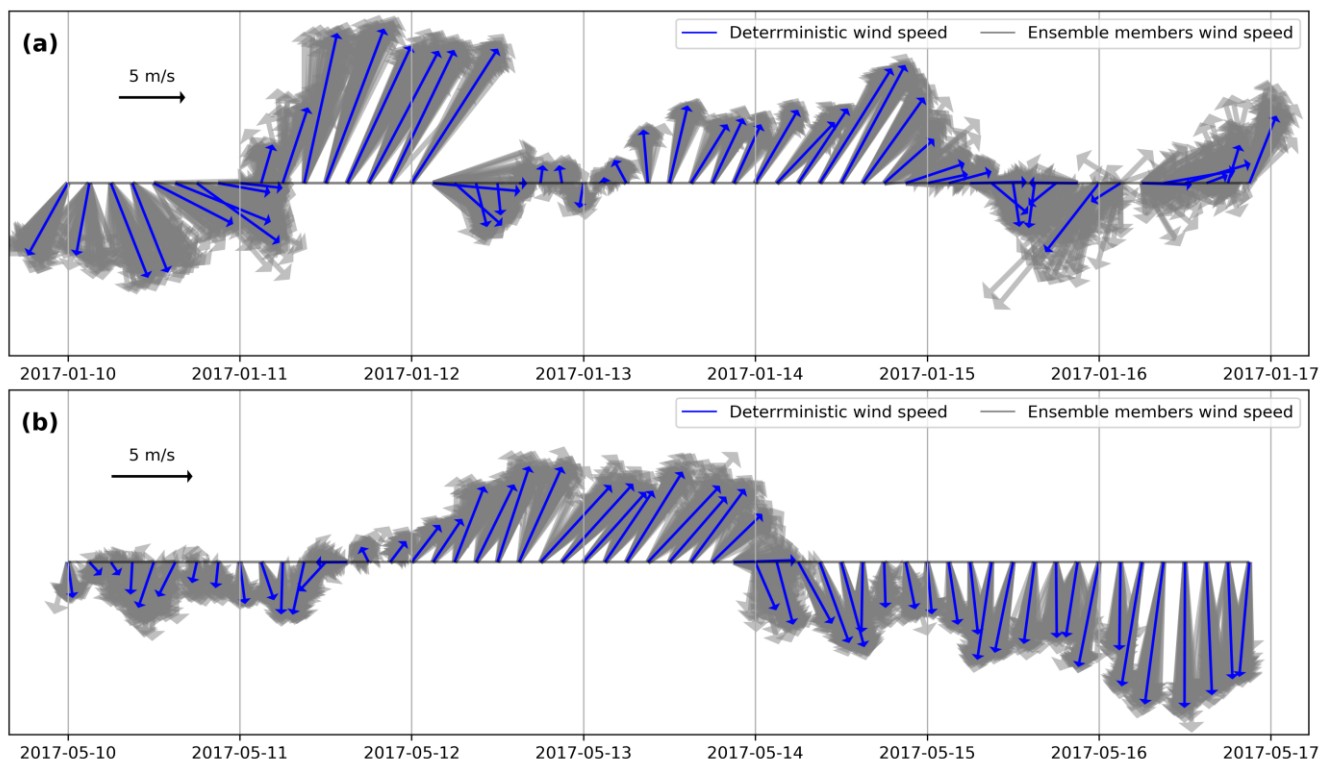

**Figure 4: Wind stick vector plots of velocity and direction at Kafireas Strait, for the 7-day deterministic simulation (blue quivers) and the 50 ensemble members (grey quivers): (a) winter, (b) spring. Reference vector 5 m/s.**

## 2.3 Oil spill metrics

### 2.3.1 Convex hull area

The convex hull of a given set of oil spill particles in the area of interest, is defined as the smallest convex polygon that contains all positions of the modelled particles. An example of the convex hull for two different sets of modelled particles, here the deterministic and one member of the ensemble, is presented in Fig. 5. In this study, the convex hull is used to examine the spreading, transport and dispersion of simulated oil spills and assess the uncertainty of the area affected by the oil particles,

considering spatial coverage differences between the ensemble and the deterministic simulation. The operational use of the convex hull, is to show the possible extent of the oil spill affecting a large area, and alert authorities to better plan for the deployment of booms for the containment of the oil spill. We should point out here that the convex hull is not by itself an uncertainty metric. We use the convex hull to introduce two more metrics and include a probabilistic information in our

prediction. The metric $A$ denotes the area of the deterministic convex hull that exceeds the area of the convex hull of an individual member selected from the ensemble, while $DA$ denotes the difference in the areas between the deterministic convex hull and the convex hull of the ensemble oil spills including all members. These two metrics are used to show the added value of the ensemble with respect to the deterministic run, as an additional information for authorities to consider polluted areas not forecasted by the deterministic approach. A schematic example of the convex hull is presented in Fig. 5, with $A$ and $DA$, being

the grey and orange hatched areas respectively. We also calculate the percentage of change of the ensemble convex hull with respect to the deterministic oil spill convex hull, in order to quantify the additional information provided by the ensemble. We define the percentage change, $a$, as:

$$a\ (\%) = \frac{DA}{\substack{Deterministic \\ convex\ hull\ area}}\ 100\%\ , \tag{6}$$

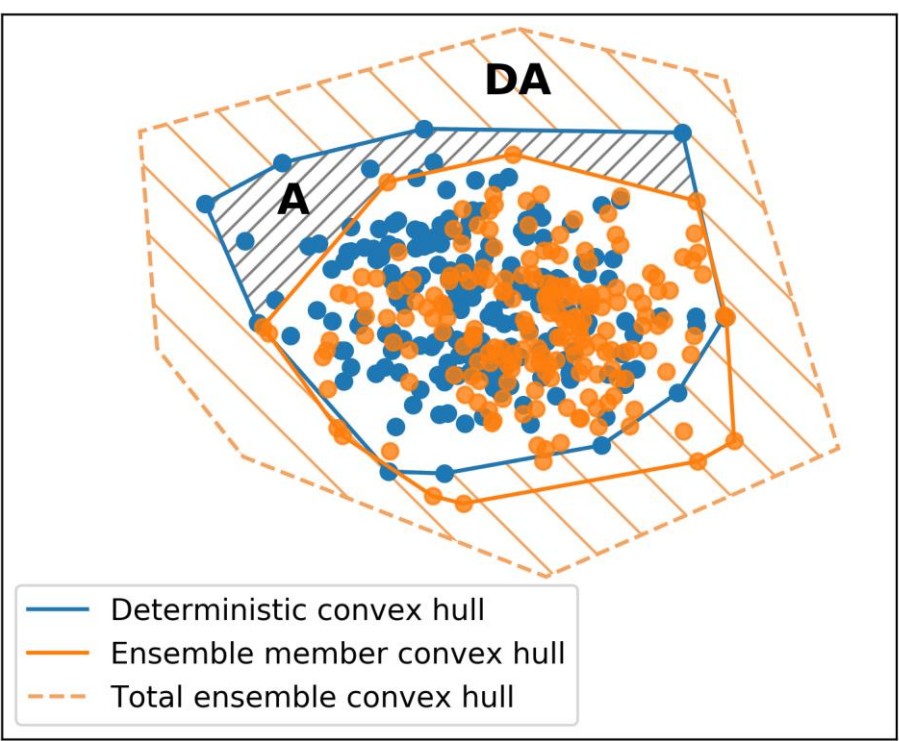

**Figure 5: Schematic of convex hull areas (blue/orange solid outer-lines) for the deterministic oil spill particles and an individual member of the ensemble (blue/orange dots). The orange dashed outer-line shows the convex hull of the whole ensemble including all members. The orange hatched area $DA$ shows the difference between the whole ensemble convex hull exceeding the deterministic**

convex hull. The grey hatched area *A* shows the difference between the deterministic convex hull exceeding the convex hull of an individual member selected from the ensemble.

## 2.3.2 Oil spill Lagrangian trajectories RMSE and uncertainty index s

We define the Lagrangian trajectory of the oil spill as the mean oil spill trajectory calculated by taking the geographical weighted mean of the released oil spill particles (and subsequently the weighted mean of the oil spill concentration) (Fig. 6). The root mean square error (RMSE) of the ensemble is estimated with respect to the deterministic simulation, calculating the separation distance between the deterministic and the ensemble oil spill Lagrangian trajectories, as a function of forecast time (De Dominicis et al., 2013b). The RMSE is given by the equation:

$$RMSE(t) = \sqrt{\frac{\sum_{n=1}^{N} D(x_e(t), x_d(t))^2}{N}}, \tag{7}$$

where $D$ is the distance between the deterministic $x_d$ and the ensemble $x_e$ Lagrangian trajectories at a given forecast time $t$ from the initial release of the particles, and $N$ is the total number of the ensemble members. According to De Dominicis et al. (2013b) and Liu and Weisberg (2011) the non-dimensional index $s$ is defined as:

$$s(t) = \frac{1}{N} \sum_{n=1}^{N} \frac{\sum_{t_0}^{t} D(x_e(t), x_d(t))}{\sum_{t_0}^{t} L(x_d(t_0), x_d(t))}, \tag{8}$$

where $D$ and $N$ have been defined in Eq. (7) and $L$ is the length of the deterministic trajectory at a given forecast time $t$ from the initial release of particles at time $t_0$. The quantities defining the $s$ index are illustrated in Fig. 6. In Eq. (8), the separation distances between the deterministic and the ensemble members are weighted by the total length of the deterministic trajectory, and it is used alongside the RMSE, as it provides a normalized index for the uncertainty quantification of the oil spill trajectories.

In most studies, the RMSE and $s$ indices are used as negative oriented metrics comparing observed and simulated trajectories to evaluate the oil spill forecast (i.e. small index values suggest good model performance). Here, we use the RMSE and $s$ index as positive oriented metrics in hypothetical accident scenarios, to quantify the added value in terms of model uncertainties using ensemble-based oil spill predictions. In this case, the ensemble provides additional information with respect to the deterministic approach, simulating several equally possible states of oil spill pollution. Unlike the convex hull which takes under account the whole area affected by the oil spill (i.e. the extent of the oil spill in distant places), the RMSE and the $s$ index metrics focus on the local conditions in close proximity to the accident area and to the heavy load of oil with the highest concentrations.

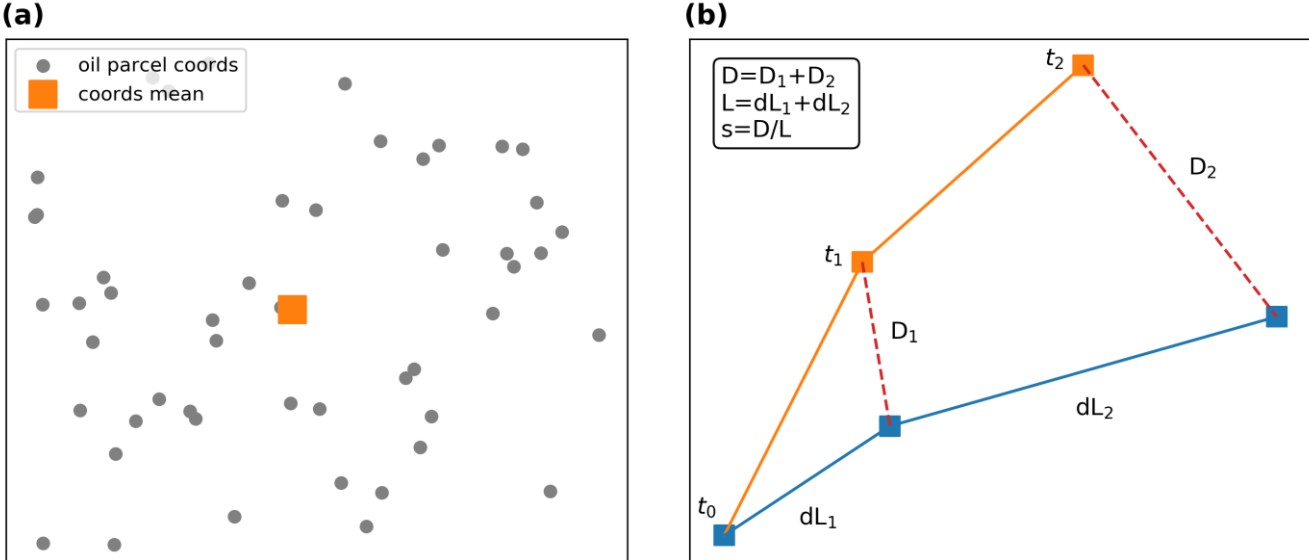

**Figure 6: Schematic of (a) spatially weighted mean (orange square) of oil spill particles (grey dots), (b) mean oil spill trajectories (following the geographically weighted mean) between the deterministic (blue line) and an individual member of the ensemble (orange line), and their corresponding distances (red dashed lines) evolving over time.**

### 2.3.3 Oiling probability

In the event of an oil spill accident, the oiling probability for a receptor (e.g. the coastline in our case) indicates the chance of the receptors' exposure to oil (Goldman et al., 2015; Amir-Heidari et al., 2019). The traditional approach for the calculation of oiling probability is based on a binary philosophy, i.e. oil spill events counted as "0" or "1" before and after the time of initial beaching respectively, regardless the amount of beached oil. Following Amir-Heidari et al. (2019), we define the oiling probability $P(t)$ as a function of the forecast time and for a total number of $N$ scenarios, as:

$$P(t) = \frac{\sum_{n=1}^{N} B_i(t)}{N},\tag{9}$$

where $B_i(t)$ takes binary values of "1" or "0" for the ith-member, at a given forecast time $t$, whether we predict oil spill in the coast or not. Here, the oiling probability can be calculated by setting (a) $N = 50$ the number of the ensemble members, indicating the percentage of how many members predict beaching, and (b) $N = 1$ for the deterministic simulation degrading the metric to a binary event, e.g. whether the deterministic simulation predicts beaching or not.

## 3 Results

### 3.1 Uncertainty assessment of oil spill spreading

We present results of oil spill accident scenarios investigating the trajectories and spreading of the deterministic and the ensemble runs, respectively. First, we examine the uncertainty metrics with respect to the convex hull, to the RMSE and to the $s$ index. The results focus mainly on the most common type of oil considering medium density API 31, which represents an intermediate case scenario with respect to the other two types of oil (e.g. heavier and lighter oil types of API 12 and API 38 respectively).

Figures 7a-c show the oil spill concentrations of the deterministic run during winter at different forecast times. The oil spill initially (Fig. 7a) spreads to the southwest due to the strong currents at Kafireas Strait and then spreads to the southeast due to changes in wind speed and direction (Figs. 7b, c). A large area in the Cyclades plateau is "polluted" with high concentrations observed near the Islands of Andros, Kea, Gyaros and Kythnos (listed from north to south). A similar broad area is also polluted using the oil spill ensemble, but there are considerable differences in the spatial distribution of the oil slicks, observed between the ensemble and the deterministic run (Fig. 7d vs Fig. 7c). Figures 7e-g show the spreading and transportation of surface oil concentrations during spring. In this experiment, the oil slick follows a different route compared to the winter simulation, initially spreading to the west and then to the north heavily polluting the coastline along the Euboea Island (Figs. 7e, f), and finally spreading southwest with the general pattern being downwind and coinciding with the sea currents exchanged through the Kafireas Strait (Figs. 7g, h). Compared with the winter period, the surface oil slick in spring is transported further to the west affecting different areas, such as the Euboea Island and reaching almost to the southeast coasts of Attika. As expected, differences in the surface oil spill distribution between the deterministic and the ensemble simulations (Fig. 7g vs Fig. 7h) are also present during spring, though being less meaningful compared with the winter period.

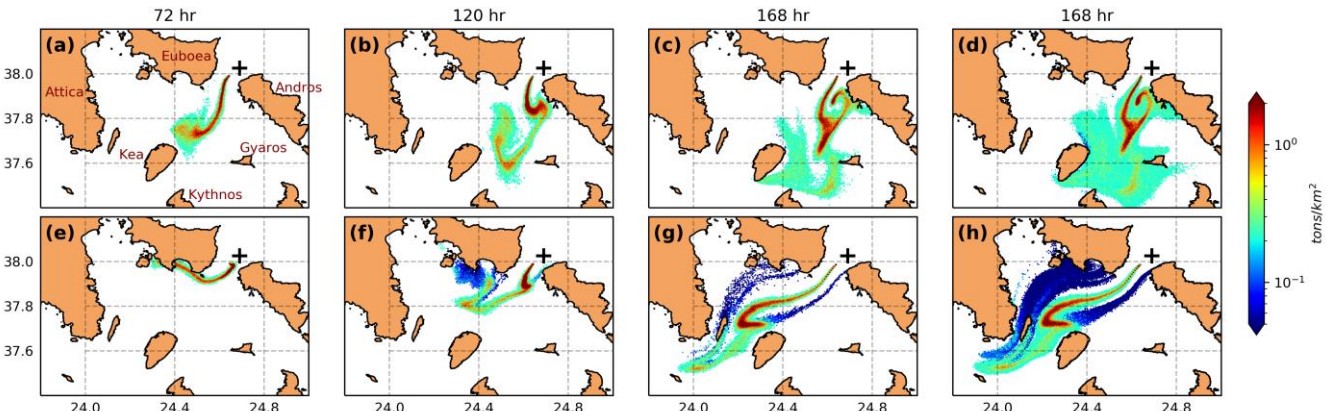

**Figure 7: Surface oil concentrations (API 31; tons/km$^2$) of the deterministic simulation for (a-c) winter and (e-g) spring, at forecast times 72, 120 and 168 hours; (d, h) same considering all members of the ensemble at forecast time 168-hour.**

Figures 8a, c show the convex hull areas of the deterministic and ensemble runs during winter, at the end of the oil spill forecast (i.e. at 168-hour forecast time). For the computation of the convex hull the following parcels were used: Figs. 8a, b, only surface parcels and parcels deposited at the coasts; Figs. 8c, d, all surface and subsurface parcels, and those deposited on the seabed and at the coasts, showing the total extent of the polluted area. Figures 8b, d show the deterministic and ensemble convex hulls during spring, with distinct but smaller differences compared with the winter period, most likely because of the lower wind intensity and the more gradual changes in wind direction (Fig. 4b). In both cases, the ensemble convex hull area is larger and fully encloses the area of the deterministic run, which is a desirable condition denoting that both approaches are consistent with each other in terms of polluted areas. Also, this highlights the added value of the ensemble with respect to the deterministic forecast, indicating a higher pollution risk for some areas predicted by particular (but equally possible) members and not predicted by the deterministic run.

In order to investigate these differences, we calculate the area of the deterministic convex hull that exceeds the convex hull area for each individual member (i.e. the $A$ metric discussed in section 2.3.1, shown in Fig. 5) as a function of the forecast time (Figs. 9a, b) taking into account all oil spill parcels (i.e. surface, subsurface and oil parcels deposited on the seabed and at the coasts). As expected, these differences in the $A$ metric gradually increase over time since the oil spill model is forced per time-step, throughout the whole simulation, with different atmospheric forcing per member. More apparent differences were observed during winter compared with spring, associated with greater wind forcing errors in that period. This is verified also by the abrupt increases in $A$ metric, observed when there are noticeable changes in wind speed and direction at specific forecast times (Fig. 4). Overall, differences between the deterministic and ensemble convex hull areas can exceed 300 km$^2$ for some members during winter, and approximately half of this area in spring, highlighting the fact that errors in wind forcing may introduce significant model uncertainties in oil spill prediction.

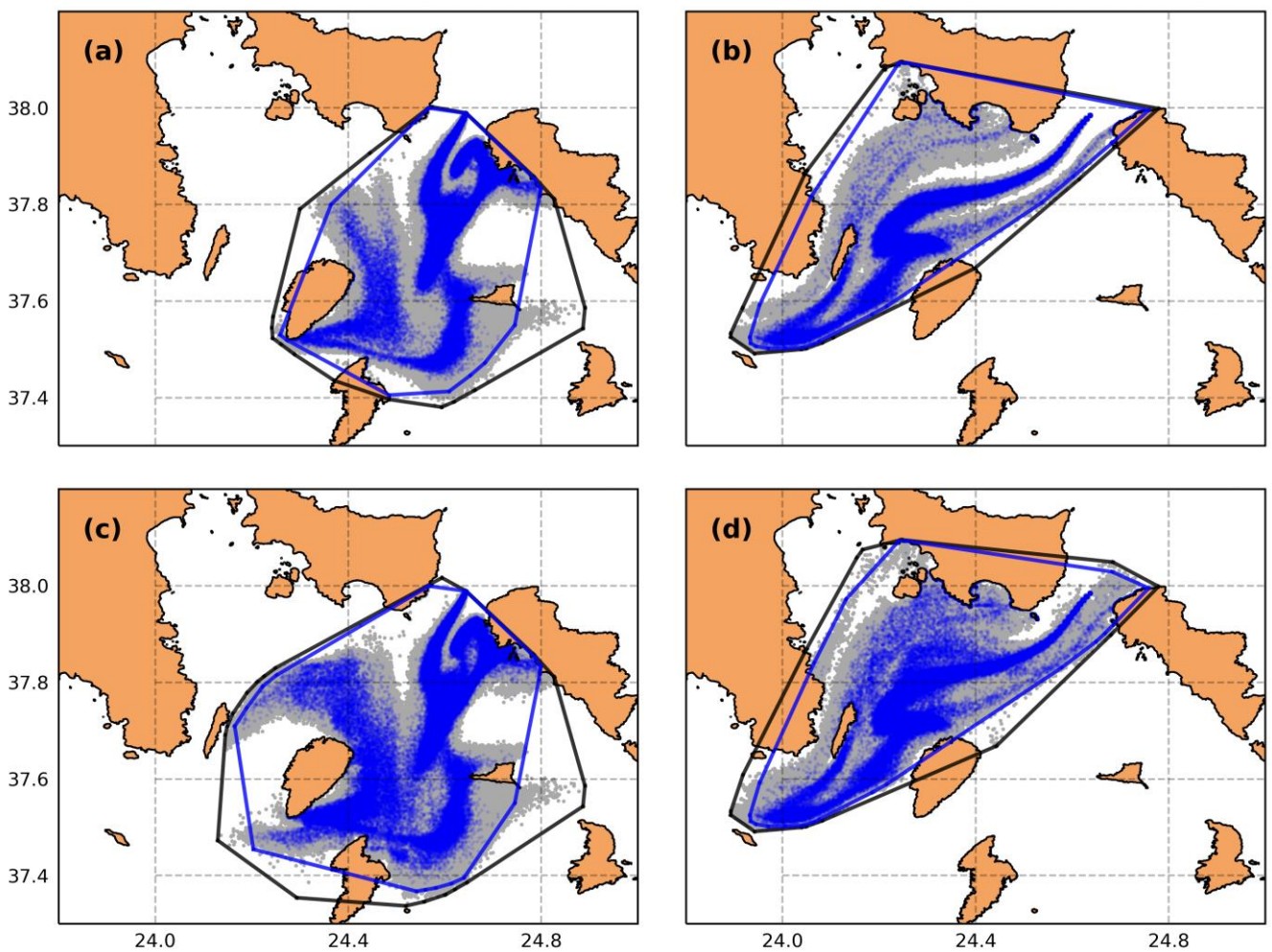

**Figure 8: Convex hulls of the deterministic (blue dots/outer-line) and all ensemble members (grey dots/outer-line), considering only surface and oil parcels deposited at the coasts (API 31), with forecast time 168-hour in (a) winter and (b) spring; (c, d) same considering all parcels (i.e. surface, subsurface and oil parcels deposited on the seabed and at the coasts).**

In addition to the $A$ metric showing one-on-one comparisons between the deterministic run and each individual member, we also calculate the $DA$ metric (shown in Fig. 5) as a function of the forecast time, denoting differences between the deterministic convex hull and the area covered by all members of the ensemble. A continuous increase is observed in the $DA$ area reaching almost up to 1.000 km$^2$ in winter (Fig. 9c) and 600 km$^2$ in spring (Fig. 9d). Although the $DA$ area is smaller during spring compared with the winter values (Fig. 9d vs. Fig. 9c), this information is just as important due to the high concentrations of beached oil (investigated in the following Section 3.2). The variability and the rate of increase are higher in the beginning of the simulation for both seasons, as denoted by the percentage of change, $a$, with respect to the deterministic hull area calculated in Eq. (6) (Figs. 9c-d). The ensemble convex hull can cover up to twice as much the area (winter period; Fig. 9c) with respect to the deterministic run in the first day of the simulation, whilst decreases to a stable percentage change at about 20% - 25%

after approximately 5-day forecast time. Overall, the percentage of additional information estimated through the ensemble than that of the deterministic convex hull, is significantly higher in the first hours of the oil spill accident and drops as the forecast time increases, pertaining to the fact that there is a continuous growth of the oil spill extent predicted by the deterministic approach (denominator in Eq. (6)). Interestingly, $a$ reaches a plateau in the diagram for both periods towards the end of the run (Figs. 9c-d).

To evaluate further the oil spill model uncertainty, we focus on the Lagrangian oil trajectories calculating the RMSE and the uncertainty index $s$ (cf. section 2.3.2). The RMSE and uncertainty index $s$ are increasing in time, denoting that the ensemble solution includes several possible states of Lagrangian trajectories that may deviate from the deterministic trajectory (Fig. 10). Higher uncertainty values for both metrics are observed during winter compared to spring, in accordance with the convex hull results. The RMSE shows also to be highly affected by wind forcing "errors", with an abrupt increase during winter and

noticeable variations during spring (Figs. 10a, b). This fact is especially true on forecast times when there are changes in wind direction among phase lagged members (Fig. 4). During spring, the RMSE does not increase monotonically and two peaks are observed around 70 and 110 hours forecast time (Fig. 10b). Apart from the wind errors explaining these variations, the high amounts of beached oil may also significantly affect the RMSE, which moderately decreases towards the end of the simulation in spring. Another remark is that results are similar in both seasons for the medium and lighter oil spill types with API 31 and

38, and only the heaviest type of oil with API 12 seems to be less impacted by the wind forcing uncertainties (Fig. 10a, b). Overall, the RMSE displays lower values during spring and for the heaviest oil type, denoting a small spread for the ensemble Lagrangian trajectories. The added value of the ensemble oil spill forecasts is shown to be more important for short temporal periods when the RMSE is increasing, suggesting that the Lagrangian trajectories of the oil spill members at these times may deviate significantly from the single trajectory of the deterministic run (at least for the medium and lighter oil types). These

RMSE abrupt variations over short periods are indicative of the wind local conditions changing rapidly, imposing an uncertainty in the oil spill prediction.

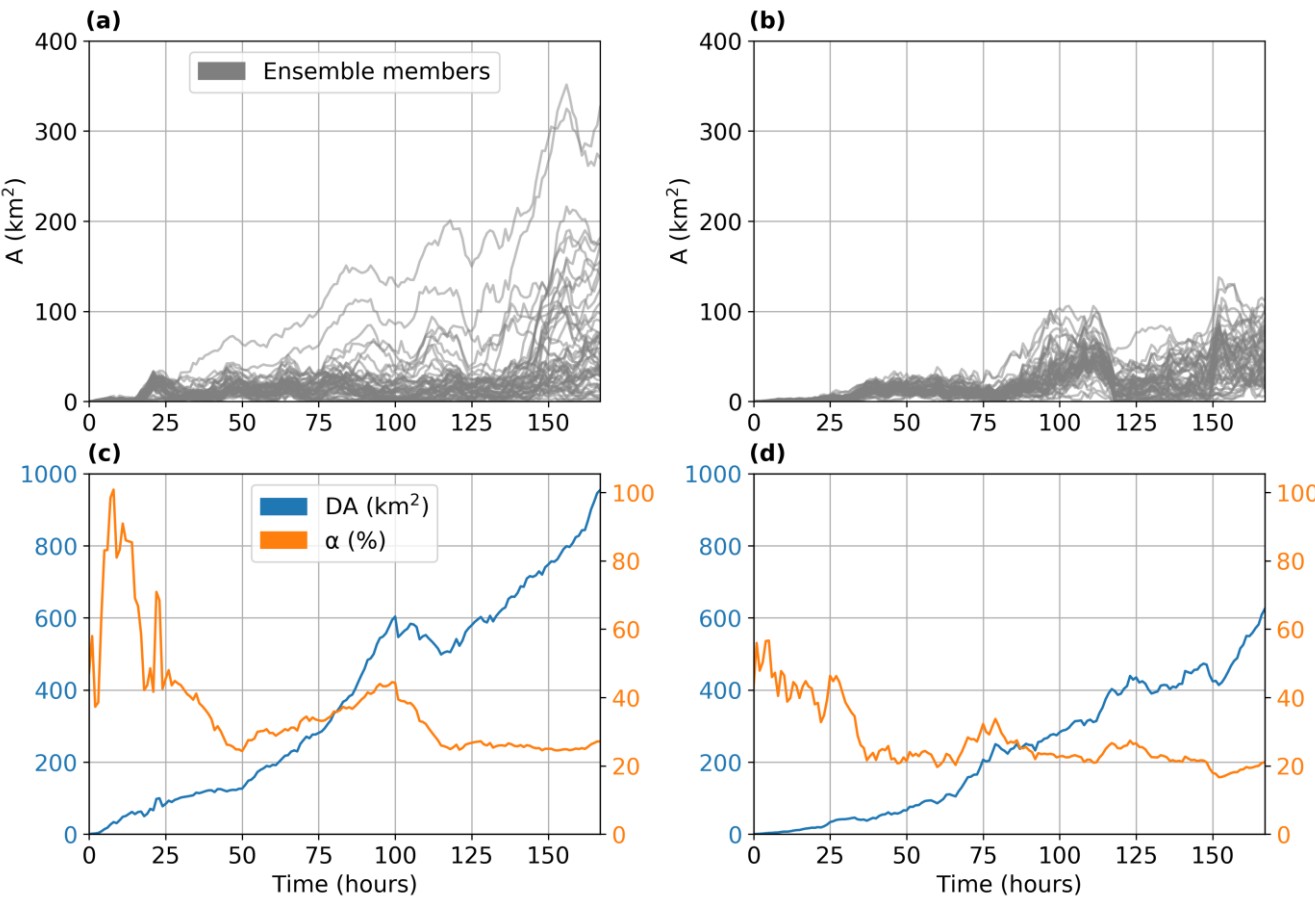

**Figure 9: Area of the deterministic convex hull that exceeds the area of each member ($A$ in km²; grey lines) for (a) winter and (b) spring, as a function of the forecast time. $DA$ extent (units in km²; blue line) of the ensemble convex hull area with respect to the deterministic state and percentage of change, $a$ (%; orange line), for (c) winter and (d) spring. Oil type API 31.**

The uncertainty index $s$ (Figs. 10c, d) shows similar information with the RMSE metric, i.e. increasing over time, but with one main difference: the $s$ index is less sensitive than the RMSE to wind forcing uncertainties. The fact that the uncertainty index $s$ is less sensitive than the RMSE, increasing at approximately constant rates, makes its use favourable in the early hours of the accident to estimate the oil spill risk and its possible evolution. For example, a higher growth rate of the oil spill trajectory expressed by the $s$ index in winter compared to spring (Figs. 10c, d), denotes a higher oil spill pollution risk that authorities may take under consideration early in the accident.

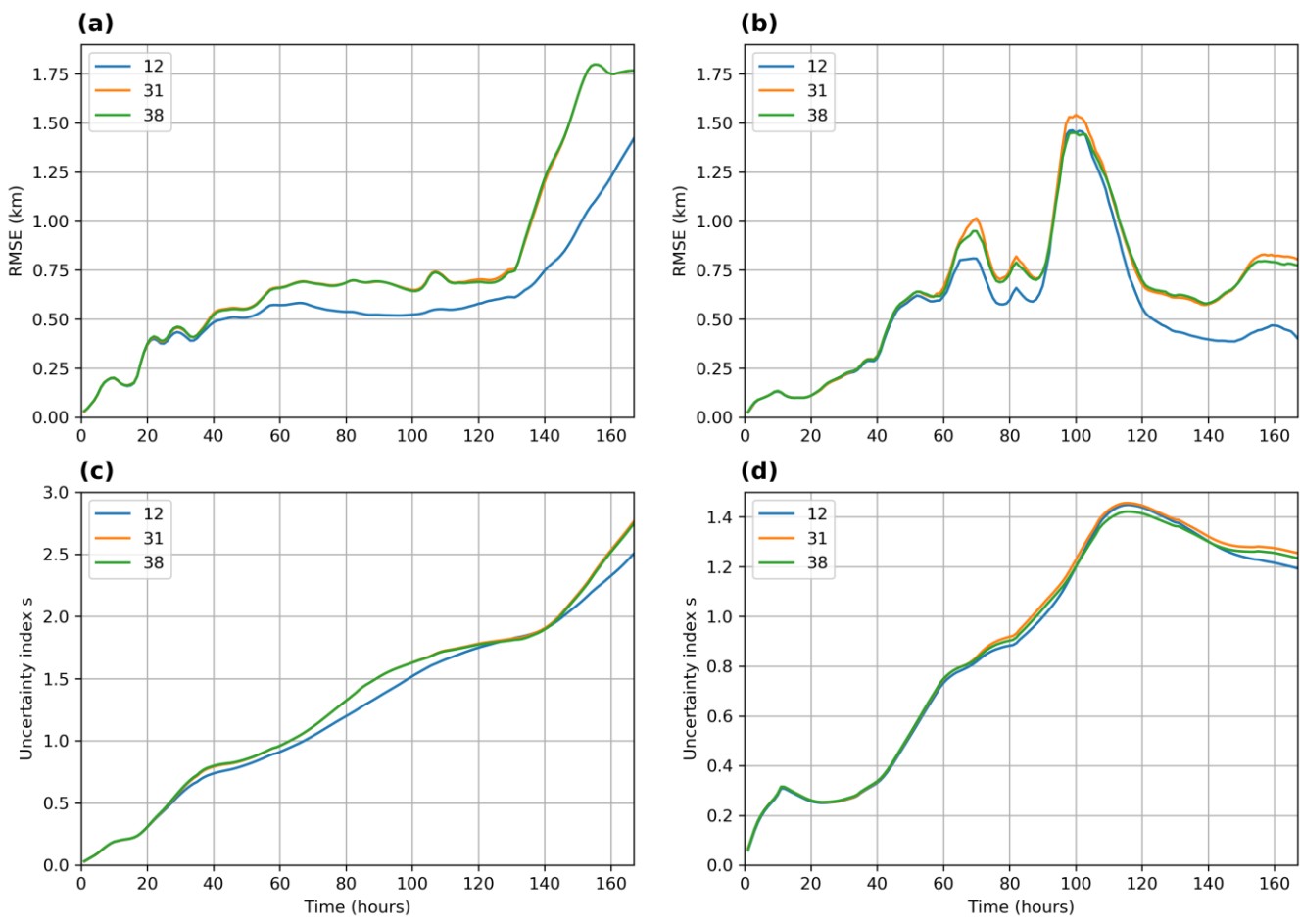

**Figure 10: RMSE (km) for (a) winter and (b) spring, as a function of the forecast time. Uncertainty index $s$ (no units) for (c) winter and (d) spring. Oil types API 12, 31 and 38.**

### 3.2 Uncertainty assessment of beached oil

"Beaching" is a term commonly used in the literature to describe the interaction between the oil and the shoreline and is an essential part of oil spill modelling and impact assessment due to the environmental, economic and social importance of coastal areas (Samaras et al., 2014). In this study, we assess the uncertainty of beached oil in the context of an oil spill ensemble, investigating equally possible states of coastal pollution. Figure 11a shows the state of beached oil concentrations during winter, at the end of the deterministic run (i.e. 168-hour forecast time). The different concentrations of beached oil particles in unit tons per coastline kilometres are presented by contrasting colours and enhanced marker sizes for better visualization of the affected areas. The deterministic simulation predicts a maximum value of 0.47 tons/km, affecting mostly the coasts in the northwestern part of Andros, the southeastern part of Euboea, as well as the Islands Kea, Kythnos and Gyaros located south of the oil spill accident (Fig. 11a). In Fig. 11b, a maximum concentration of 1.44 tons/km is predicted by the ensemble, with marked spatial differences against the deterministic state. During spring, a larger amount of beached oil is observed with hit

locations in Euboea and Andros Islands, and maximum values at about 13.52 tons/km for the deterministic run (Fig. 11c) and 30.31 tons/km for the ensemble (Fig. 11d). This fact alone indicates the high degree of uncertainty in the amount of beached oil and the coastal pollution predicted by a single oil spill state.

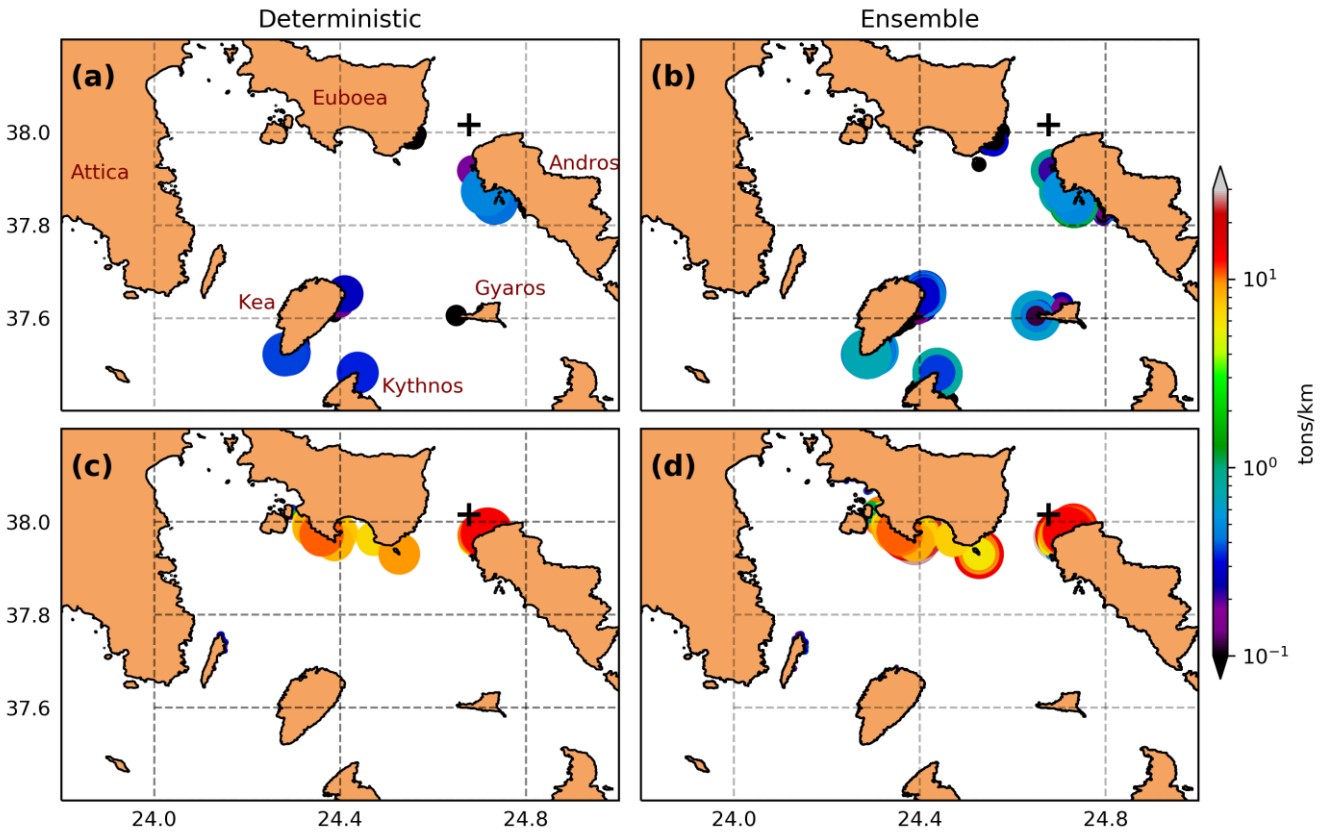

**Figure 11: Beached oil concentrations (API 31; tons/km) at the end of the run, i.e. forecast time 168-hour, during (a-b) winter and (c-d) spring, for the (a, c) deterministic and the (b, d) ensemble considering all members.**

The oiling probability metric is based on a binary approach and informs about the arrival time of fixed oil in the coast as predicted by the model. In the context of an ensemble, this can be used to infer uncertainties in the hit time for beached oil. The oiling probability of a determinist state is depicted in the form of a Heaviside step function going from zero to one at hit
time (e.g. dashed lines in Fig. 12). This binary representation is also true per individual member, but we can show this information considering the whole ensemble and expressing the event of beached oil in a probabilistic way of a cumulative density function. For instance, during winter, we can show that the model uncertainty of beached oil for type API 31 is expressed in the form of a temporal window for all ensemble members and has a 15-hour duration between 20 and 35 hours forecast time (Fig. 12a). This practically means that before the 20-hour time mark the oiling probability of total fixed oil on
the coast is 0% (i.e. none of the members predict beached oil) and after the 35-hour time mark is 100% (i.e. all members predict beached oil from this time and on). Between these two time marks the cumulative probability of the ensemble increases,

suggesting that the number of members predicting beached oil gradually increases. During spring, the uncertainty of the hit time as predicted by the ensemble, is shown over a shorter temporal window (i.e. at about 8 hours, between 32 and 40 hours forecast time). An interesting fact between the deterministic and the ensemble hit times, is that in almost every occasion the deterministic beached oil is predicted within the temporal window provided by the ensemble. Overall, this information of cumulative binary events between members can be of added value as opposed to a single hit time prediction.

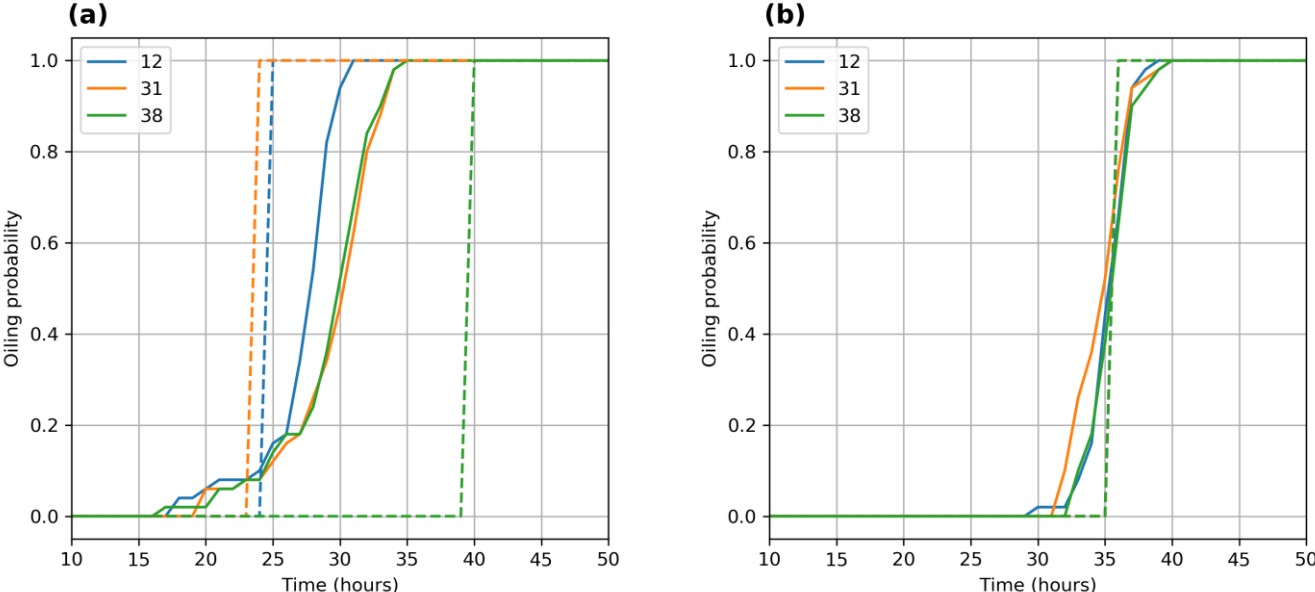

Figure 12: Oiling probability P (no units) for (a) winter and (b) spring, as a function of the forecast time. Different colours denote oil spill types API 12, 31 and 38. Dashed lines for the deterministic state and solid lines for the ensemble.

### 3.3 Uncertainty assessment of weathering processes

Oil spill parcels undergo modifications due to fate processes modelled by the weathering transformation Eq. (2). A fraction of the oil spill evaporates, mainly depending on wind speed and sea-surface temperature (other processes include vapor pressure, thickness etc.), and the rest emulsify i.e. absorb water altering its intrinsic properties, such as viscosity and volume. Evaporation and emulsification increase viscosity, whereas the oil spill volume changes with dispersion, for example, the water column oil uptake is enhanced by waves and in shallow areas can become sedimented on the seabed. The process of oil at the beach may not be permanent, since oil particles in some occasions may be washed back from the coasts to the seawater.

In this section, we discuss the model uncertainties of the above-mentioned oil weathering processes, in relation to wind forcing uncertainties. Figure 13 shows the temporal evolution of the main fate parameters derived from the deterministic and the oil spill ensemble. The oil mass conservation law, at each time-step, demands that the modelled total oil concentration equals the surface, evaporated, dispersed (including sedimented) and total oil at the coasts (including both fixed and free oil at the coast) (Figs. 13a, b). Evaporation processes are almost constant with small variations from the early hours of the spill, and exhibit

negligible model uncertainties throughout the whole run (Figs. 13a, b). Noticeable changes are observed in the surface oil. As expected, surface oil is decreased in time, compensated by the increased dispersion and beaching processes (Figs. 13a, b). Surface oil uncertainties are almost 10% of the total oil concentrations towards the end of the ensemble runs. This is also valid for the dispersion and total coastal oil, where dispersion shows higher model uncertainties during the winter ensemble and total coastal oil during the spring ensemble. Emulsification model uncertainties appear to be significant in the first hours of the accident when surface oil has not declined yet, and dispersion and beaching processes are still small (Figs. 13c, d). The emulsion viscosity reaches its maximum value and becomes constant when dispersion and beaching processes start to develop. The volume ratio of water over oil (i.e. water/oil) shows significant variations and noticeable model uncertainties throughout the whole ensemble, though its values decrease towards the end of the run, possibly because of the surface oil decline (Figs. 13c, d). In case of heavier oil type (API 12; not shown) evaporation and emulsion viscosity are lower compared with the values shown in Fig. 13 (i.e. API 31), while dispersed and total coastal oil exhibit higher percentages. The opposite is true in case of lighter oil type (API 38; not shown), but with small differences. Surface oil and volume ratio are higher in the first hours of the accident for the heavier oil type (API 12; not shown) compared with the values shown in Fig. 13 (i.e. API 31), and lower towards the end of the run.

Overall, the model uncertainties of the oil weathering processes appear to be moderate, pertaining to the fact that: (a) many factors influencing these processes remain unchanged across the ensemble members (e.g. sea-surface temperature, vapor pressure etc.), (b) the wind speed, being an important factor for the control of the fate parameters, is not very different between the ensemble mean and the deterministic wind speed, and (c) the phase errors introduced by time-lagged members in the wind direction, are mainly important for the spreading of the oil spill trajectories and only moderately important for the estimation of the weathering processes. In light of these findings, additional error processes to the wind forcing should be envisaged to increase weathering model uncertainties. In an operational context, the provided information regarding the fate parameters can be potentially important to better plan methods of treatment, e.g. the spraying of surfactants on the oil slick based on model uncertainties for the emulsion viscosity.

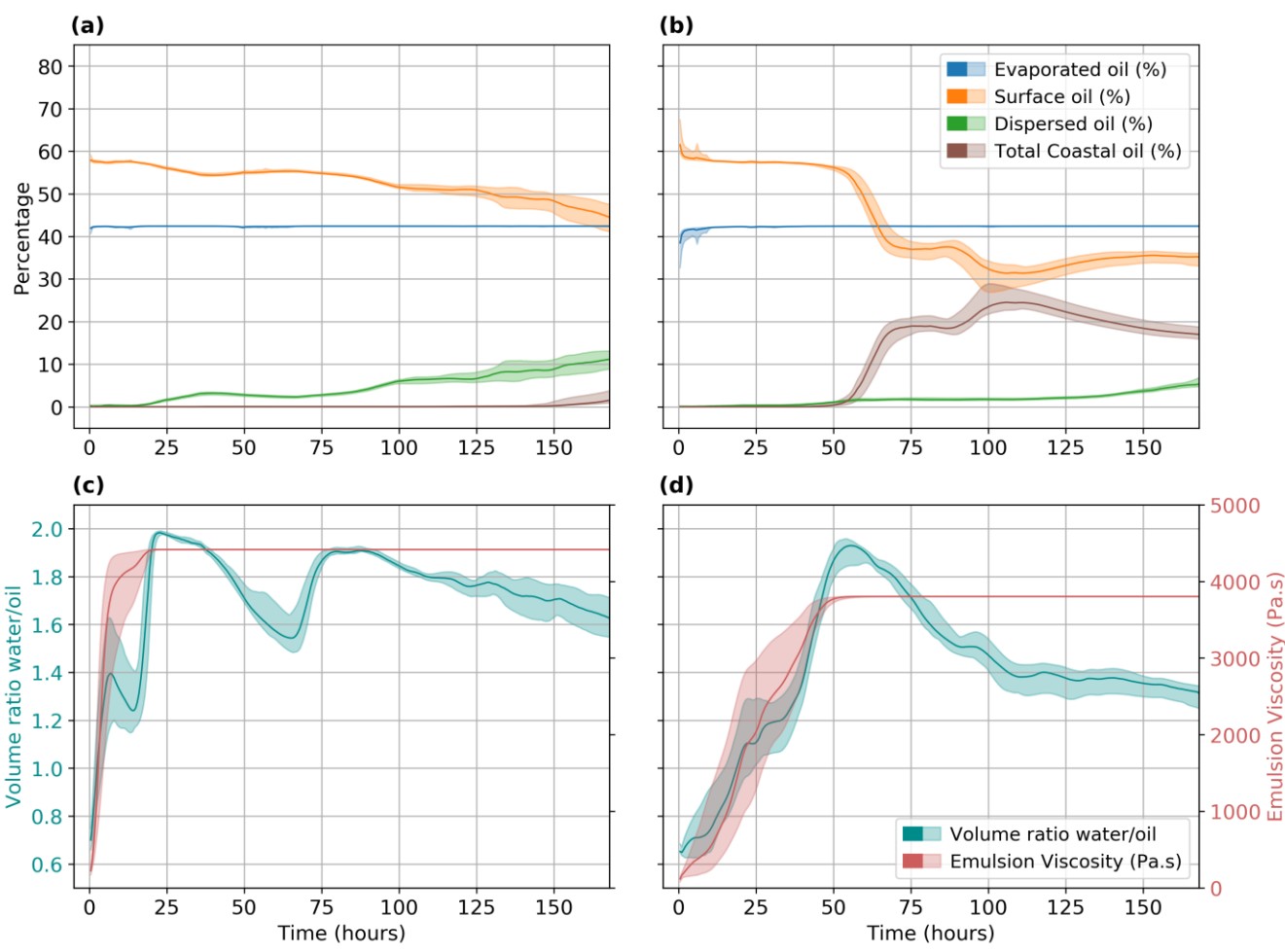

**Figure 13: Fate parameters. Percentages (%) of evaporated, surface, dispersed (including sedimentation) and total coastal oil for (a) winter and (b) spring, as a function of the forecast time. Volume ratio water/oil and emulsion viscosity (Pa.s) for (c) winter and (d) spring. Coloured solid lines for the deterministic run and coloured shaded areas for the ensemble. Oil type API 31.**

## 4 Conclusions

The study aims at evaluating the impact of atmospheric forcing uncertainties on the performance of oil spill model prediction and dispersion of pollutants in the marine environment. We performed an ensemble of oil spill simulations, using an ensemble of wind forcings from the ECMWF ensemble prediction system. The atmospheric forcing was used to generate oil spill model uncertainties in a regional domain of the Aegean Sea, carried out with the model MEDSLIK-II. We investigated model uncertainties based on the spreading, transport, and the extent of the oil spill, including surface, subsurface and oil particles

deposited on the seabed and at the coasts. We also investigated model uncertainties for the hit time and location of beached oil. The goal is to ascertain whether the information provided by the oil spill ensemble is important with respect to the

deterministic run, and if an atmospheric ensemble can be used to improve oil spill probabilistic prediction, increasing the reliability of the prediction.

An atmospheric ensemble of 50 members was used for the oil spill model forcing, for two different seasons (i.e. winter and spring) and three different types of oil, performing 7-day simulations. The results indicated that the wind forcing greatly influenced the oil spill dispersion in the region, being important for the model performance in nearshore areas. The dispersion pattern among ensemble members was in the same general direction as in the deterministic approach, but there were considerable variations in the transport, evolution, shape and size in the oil spill forecasts. Model uncertainties were more meaningful for highly variable forcing patterns, with abrupt changes in wind direction and intensity.

The extent of the polluted area predicted by the oil spill ensemble was found to be greater than the area predicted by the deterministic simulation by 20% - 100%. This additional information was verified by the use of the convex hull and its associated probabilistic metrics, and was more important in the first hours of the oil spill accident. Depending on the season and the type of oil, the continuous growth of the oil spill extent predicted by the model ensemble can be potentially important to monitor pollution and promote strategies of response. In addition, uncertainty estimates derived by the RMSE can be used in an ensemble protocol, alongside a deterministic run, to show model uncertainties of abrupt changes in oil spill trajectories (here due to wind forcing uncertainties) alerting authorities to operate in a narrow temporal window. On the other hand, the $s$-index is less sensitive to model uncertainties because it is a normalized index, and can be used to assess the evolution of the oil spill early in the accident, with moderate dependence on wind forcing errors over the period under investigation. The model uncertainties, can also provide us with important information about the concentrations, hit locations and hit time of beached oil, evaluating the impact on the coastal environment and better planning for a number of equally possible pollution scenarios. In general, for highly variable wind forcing fields, the uncertainty generated by the atmospheric ensemble appears to be more important for the lighter types of oil (i.e. high API values) mostly in the open ocean. However, even for less variable wind fields, the atmospheric ensemble is able to provide meaningful information in high polluted coastal areas with the amount of beached oil becoming important mostly for the heavier oil types (i.e. lower API values). Finally, model uncertainties of the oil spill fate parameters were found to be moderate and additional error processes to the wind forcing should be envisaged to increase weathering model uncertainties.

An oil spill ensemble prediction system, based on wind forcing uncertainties, can be useful predicting equally possible oil spill states, that are more informative compared with the deterministic run as the forecast is extended in time. As a concluding remark, the ensemble forecasts show great potential to improve the reliability of an oil spill prediction and be used operationally as an important tool, to better plan and direct the available resources for the control and mitigation procedures, in the event of an oil spill.

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
