# Peer review of "Oil spill model uncertainty quantification using an atmospheric ensemble"

_Ocean Science, 2021_

## Author Comment (AC1)

**Response to reviewer #1**

The answers to the interactive comments by reviewer #1 have been shared with the co-authors of the manuscript.

**Reviewer #1**

The ms is of very interest in terms of examining new approaches for the improvement of the oil spill predictions due to surface forcing uncertainties.

The use of the ECMWF ensemble wind forecast can provide valuable information to the response agencies regarding the impacted area, both at sea and at shoreline, as well as of the variation of the weathering processes.

**Author**

We thank the reviewer for the constructive and positive comments. In the revised manuscript, we will address all reviewer's comments.

**Reviewer #1**

Minor comments:

Clarify the temporal resolution of the used ECMWF deterministic and ensemble wind forecasting data, as from the Figure 4 it seems that are 3 hourly? Similarly, provide the temporal resolution of the used CMEMS Med MFC sea currents.

**Author**

Thank you for the careful reading, the deterministic and ensemble wind forcing were retrieved by ECMWF archives at 3-hour frequency. The Med MFC data were provided by CMEMS at daily frequency. For both datasets, we used the MEDSLIK-II pre-processing tools to generate hourly metocean forcing fields as required by the oil spill model.

This information will be included in Section 2.2 of the revised manuscript: "converting the oil spill model inputs from the CMEMS daily ocean forcing and the ECMWF 3-hour atmospheric forcing to hourly fields."

**Reviewer #1**

Clarify if the DA is applied also to the southern part of the convex hull area, as from Figure 5 the DA it seems that it was applied only in the northern part.

**Author**

We apologize to the reviewer for the confusion depicting the convex hull areas.

In Figure 5, we show the deterministic convex hull (blue) and the convex hull of an individual member (orange) selected from the ensemble. We focus on the hatched area in the northern part, where the deterministic convex hull exceeds the convex hull of the individual member. This area is named in the ms with the letter "A". In order to keep the schematic as simple as possible, we do not depict the area "DA" denoting the difference between the deterministic convex hull of all members of the ensemble (not shown).

Figure 5 will be revised to better indicate the hatched area "A". Also, we will explain better in the revised text the hatched area "A".

**Reviewer #1**

The figure 8 to be renamed as figure 8a and add an additional map as figure 8b showing the extend of the deterministic and ensemble spill extend on the sea surface and on shoreline only. The transport of the dispersed oil, i.e. the transport of the subsurface oil in oil spill modeling is calculated using sea currents, not winds.

**Author**

We agree with the reviewer that the transport of the subsurface oil is calculated using the sea currents.

Following the reviewer's suggestion, we will rename Figs. 8a, b as Figs. 8c, d, and we will add two additional maps as new Figs. 8a, b, showing the deterministic and ensemble spill extent on the sea surface and on the shoreline, during winter and spring respectively. Also, we will revise the text to include the aforementioned changes.

**Reviewer #1**

It will be an added value of the ms the provision of the plots of the main weathering parameters derived from the deterministic oil spill simulation and those derived from the ensemble oil spill simulations (mean averaged).

**Author**

We thank the reviewer for the insightful comment. In our application, the differences between the deterministic and ensemble simulations were mainly attributable to phase errors in the wind direction controlling advection and oil spill trajectories. The oil spill state due to weathering processes showed relatively small differences between the deterministic and ensemble simulations.

Following the reviewer's suggestion, we will revise the text and provide information for the ensemble mean and spread (i.e., 1std) for weathering parameters (e.g., water/oil volume ratio, emulsion viscosity, emulsion density, evaporative volume).

**Reviewer #1**

For operational oil spill predictions is of interest to the response agencies to provide the run time required for the deterministic oil spill prediction and of the run time required for the ensemble oil spill predictions for 48 hours, 72hours, 120hours and 240hours.

**Author**

Thank you, we agree that for operational predictions the computational run time is of particular interest to the response agencies.

The text will be revised accordingly: "The run time of our simulations was mainly determined by the number of oil parcels and the size of the ensemble. For a 168-hour (7-day) oil spill prediction (in our domain of interest depicted in Fig. 1), the deterministic simulation required approximately a 20-minute run time, including the model's I/O tasks. The computational cost of the ensemble prediction, in case all members are run in serial mode (i.e., one after the other in sequence), is analogous to the number of the ensemble members. The latter is valid also for the data storage. For a small ensemble simulation in a HPC facility with available CPU cores, the ensemble can run in parallel mode (i.e., the members can run independently) and the computational cost will be the same as the deterministic simulation."

---

## Author Comment (AC2)

**Response to reviewer #2**

The answers to the interactive comments by reviewer #2 have been shared with the co-authors of the manuscript.

**Reviewer #2**

In the present manuscript the Impact of using an ensamble atmospheric forcing on a oil trajectory and wheathering model is studied.

I Found the paper well written and focused on a relevant argument on which it shed some light.

The paper does not clarify how much the approach can improve the solution, while it observes an increase of possible oil beaching (20 to 100 percent more than the deterministic solution). Anyway, i found that the paper deserves to be published.

**Author**

We thank the reviewer for the constructive and positive comments. In the revised manuscript, we will address the reviewer's main suggestion. Following also the suggestion from another reviewer, we will add information for some variables of the ensemble simulation.

**Reviewer #2**

I just suggest an inmprovement of section 2 with a more detailed description of the differences in the implementation of the ensamble vs deterministic simulation. In particular it is not clear to me the approach used in simulating with the ensamble solution. Is it used the "ensamble" averaged solution or the members of the ensamble are treated as single runs? In other words does the oil spill model is ran 50 times and then actually an ensamble oil spill trajectory and evolution of oil is considered? By reading "ensamble oil spill model" I would be induced to figure out an actual ensamble of trajectory, but it is unclear to me if Authors actually performed an ensamble of trajectory. In negative case, i.e. if authors just ran a "deterministic" oil spill by using the averaged solution of an ensamble atmospheric forcing, I would suggest to revise the text rewording sentences relative to "oil spill ensamble".

**Author**

We apologize for the confusion. The reviewer is correct to note that by reading the phrase *"ensemble oil spill model"* an ensemble of 50 simulations should be expected.

In this study, we have performed exactly what the reviewer anticipates, i.e., an ensemble of 50 simulations, where each oil spill member uses different atmospheric forcing obtained from the ECMWF ensemble prediction system.

We have not estimated the average of the atmospheric ensemble to force the oil spill model, because this would result to a "virtual" mean atmospheric forcing and the oil spill results would be less meaningful compared to the approach followed in this work.

We will revise the text in Section 2 to better clarify the ensemble approach.

---

## Author Comment (AC3)

**Response to reviewer #1**

The answers to the interactive comments by reviewer #1 have been shared with the co-authors of the manuscript.

At the end of the responses we attach a "tracked changes" version of the manuscript. The new lines in the responses refer to the revised ms. with accepted the track changes.

**Reviewer #1**

The ms is of very interest in terms of examining new approaches for the improvement of the oil spill predictions due to surface forcing uncertainties.

The use of the ECMWF ensemble wind forecast can provide valuable information to the response agencies regarding the impacted area, both at sea and at shoreline, as well as of the variation of the weathering processes.

**Author**

We thank the reviewer for the constructive and positive comments. In the revised manuscript, we have addressed all reviewer's comments.

**Reviewer #1**

Minor comments:

Clarify the temporal resolution of the used ECMWF deterministic and ensemble wind forecasting data, as from the Figure 4 it seems that are 3 hourly? Similarly, provide the temporal resolution of the used CMEMS Med MFC sea currents.

**Author**

Thank you for the careful reading, the deterministic and ensemble wind forcing were retrieved by ECMWF archives at 3-hour frequency. The Med MFC data were provided by CMEMS at daily frequency. For both datasets, we used the MEDSLIK-II pre-processing tools to generate hourly metocean forcing fields as required by the oil spill model.

This information is now included in Section 2.2 of the revised manuscript (new lines 140-141).

**Reviewer #1**

Clarify if the DA is applied also to the southern part of the convex hull area, as from Figure 5 the DA it seems that it was applied only in the northern part.

**Author**

We apologize to the reviewer for the confusion depicting the convex hull areas.

In the revised Figure 5, we show the deterministic convex hull (blue solid outer-line), the convex hull of an individual member selected from the ensemble (orange solid outer-line), and the total ensemble convex hull including all members (orange dashed outer-line). We focus on the hatched area in the northern part (i.e. grey parallel lines), where the deterministic convex hull exceeds the convex hull of the individual member. This area is named in the ms. with the letter "A", now included in the revised figure. In addition, we show in the revised figure the hatched area named in the ms. "DA" (i.e. orange parallel lines), depicted as the area where the convex hull of the whole ensemble exceeds the deterministic convex hull.

We have revised the ms. to better explain the areas named "A" and "DA", as well as the caption in the revised Fig. 5 (new lines 186-190 and 273, 287).

**Reviewer #1**

The figure 8 to be renamed as figure 8a and add an additional map as figure 8b showing the extend of the deterministic and ensemble spill extend on the sea surface and on shoreline only. The transport of the dispersed oil, i.e. the transport of the subsurface oil in oil spill modeling is calculated using sea currents, not winds.

**Author**

We agree with the reviewer that the transport of the subsurface oil is calculated using the sea currents.

Following the reviewer's suggestion, we renamed Figs. 8a, b as Figs. 8c, d, and we added two additional maps as new Figs. 8a, b, showing the deterministic and ensemble spill extent on the sea surface and on the shoreline, during winter and spring respectively.

We have revised the text to include the aforementioned changes (new lines 262-265 and 274-275).

**Reviewer #1**

It will be an added value of the ms the provision of the plots of the main weathering parameters derived from the deterministic oil spill simulation and those derived from the ensemble oil spill simulations (mean averaged).

**Author**

We thank the reviewer for the insightful comment. In our application, the differences between the deterministic and ensemble simulations were mainly attributable to phase errors in the wind direction controlling advection and oil spill trajectories. The oil spill state due to weathering processes showed moderate differences between the deterministic and ensemble simulations.

Following the reviewer's suggestion, we added a new section in the revised ms. entitled "3.3 Uncertainty assessment of weathering processes" discussing the model uncertainties of the oil weathering processes, in relation to wind forcing uncertainties (new lines 365-399).

A new Fig. 13 is also inserted in this section, showing the main weathering parameters derived from the deterministic and the ensemble simulations.

We also revised the Conclusions Section 4 (new lines 434-436).

**Reviewer #1**

For operational oil spill predictions is of interest to the response agencies to provide the run time required for the deterministic oil spill prediction and of the run time required for the ensemble oil spill predictions for 48 hours, 72hours, 120hours and 240hours.

**Author**

Thank you, we agree that for operational predictions the computational run time is of particular interest to the response agencies.

We have revised accordingly the ms. to include this information (new lines 123-129).

[revised manuscript text omitted]

---

## Author Comment (AC4)

**Response to reviewer #2**

The answers to the interactive comments by reviewer #2 have been shared with the co-authors of the manuscript.

At the end of the responses we attach a "tracked changes" version of the manuscript. The new lines in the responses refer to the revised ms. with accepted the track changes.

**Reviewer #2**

In the present manuscript the Impact of using an ensamble atmospheric forcing on a oil trajectory and wheathering model is studied.

I Found the paper well written and focused on a relevant argument on which it shed some light.

The paper does not clarify how much the approach can improve the solution, while it observes an increase of possible oil beaching (20 to 100 percent more than the deterministic solution). Anyway, i found that the paper deserves to be published.

**Author**

We thank the reviewer for the constructive and positive comments. In the revised manuscript, we have addressed the reviewer's main suggestions.

Following also the suggestion from another reviewer, we added information for some variables of the ensemble simulation (i.e. the main weathering processes, cf. new Section 3.3 and new Fig. 13) providing more information that could potentially be important in an operational context and improve the solution.

**Reviewer #2**

I just suggest an inmprovement of section 2 with a more detailed description of the differences in the implementation of the ensamble vs deterministic simulation. In particular it is not clear to me the approach used in simulating with the ensamble solution. Is it used the "ensamble" averaged solution or the members of the ensamble are treated as single runs? In other words does the oil spill model is ran 50 times and then actually an ensamble oil spill trajectory and evolution of oil is considered? By reading "ensamble oil spill model" I would be induced to figure out an actual ensamble of trajectory, but it is unclear to me if Authors actually performed an ensamble of trajectory. In negative case, i.e. if authors just ran a "deterministic" oil spill by using the averaged solution of an ensamble atmospheric forcing, I would suggest to revise the text rewording sentences relative to "oil spill ensamble".

**Author**

We apologize for the confusion. The reviewer is correct to note that by reading the phrase *"ensemble oil spill model"* an ensemble of 50 simulations should be expected.

In this study, we have performed exactly what the reviewer anticipates, i.e., an ensemble of 50 simulations, where each oil spill member uses different atmospheric forcing obtained from the ECMWF ensemble prediction system.

We have not estimated the average of the atmospheric ensemble to force the oil spill model, because this would result to a "virtual" mean atmospheric forcing and the oil spill results would be less meaningful compared to the approach followed in this work.

The new lines 112-115 in Section 2 were added to better clarify the ensemble approach.

[revised manuscript text omitted]